# The Central Contributions of Breast Cancer Stem Cells in Developing Resistance to Endocrine Therapy in Estrogen Receptor (ER)-Positive Breast Cancer

**DOI:** 10.3390/cancers11071028

**Published:** 2019-07-22

**Authors:** David Rodriguez, Marc Ramkairsingh, Xiaozeng Lin, Anil Kapoor, Pierre Major, Damu Tang

**Affiliations:** 1Department of Medicine, McMaster University, Hamilton, ON L8S 4K1, Canada; 2The Research Institute of St Joe’s Hamilton, St Joseph’s Hospital, Hamilton, ON L8N 4A6, Canada; 3Urological Cancer Center for Research and Innovation (UCCRI), St Joseph’s Hospital, Hamilton, ON L8N 4A6, Canada; 4The Hamilton Center for Kidney Research, St. Joseph’s Hospital, Hamilton, ON L8N 4A6, Canada; 5Department of Surgery, McMaster University, Hamilton, Hamilton, ON L8S 4K1, Canada; 6Division of Medical Oncology, Department of Oncology, McMaster University, Hamilton, ON, L8V 5C2, Canada

**Keywords:** ER-positive breast cancer, endocrine therapy resistance, breast cancer stem cells, hormone and growth factor signaling, microenvironment

## Abstract

Breast cancer stem cells (BCSC) play critical roles in the acquisition of resistance to endocrine therapy in estrogen receptor (ER)-positive (ER + ve) breast cancer (BC). The resistance results from complex alterations involving ER, growth factor receptors, NOTCH, Wnt/β-catenin, hedgehog, YAP/TAZ, and the tumor microenvironment. These mechanisms are likely converged on regulating BCSCs, which then drive the development of endocrine therapy resistance. In this regard, hormone therapies enrich BCSCs in ER + ve BCs under both pre-clinical and clinical settings along with upregulation of the core components of “stemness” transcriptional factors including SOX2, NANOG, and OCT4. SOX2 initiates a set of reactions involving SOX9, Wnt, FXY3D, and Src tyrosine kinase; these reactions stimulate BCSCs and contribute to endocrine resistance. The central contributions of BCSCs to endocrine resistance regulated by complex mechanisms offer a unified strategy to counter the resistance. ER + ve BCs constitute approximately 75% of BCs to which hormone therapy is the major therapeutic approach. Likewise, resistance to endocrine therapy remains the major challenge in the management of patients with ER + ve BC. In this review we will discuss evidence supporting a central role of BCSCs in developing endocrine resistance and outline the strategy of targeting BCSCs to reduce hormone therapy resistance.

## 1. Introduction

Approximately 1.7 million women are diagnosed with breast cancer (BC) every year; the disease causes half a million deaths annually worldwide [1,2]. BC is a heterogeneous disease; it consists of tumors expressing estrogen receptor (ER) and/or HER2, as well as tumors negative for ER, progesterone receptor (PR), and HER2 expression (triple negative/TN), with approximately 75% of cases being ER-positive (ER + ve) [3,4,5]. This is consistent with an essential role of ER in breast development; in humans, ER expression was detected in breast epithelial cells from 30 weeks gestation and onward [6]; in mice deficient of ER, the rudimentary ductal structures of early gestation were unable to develop [7]. Likewise, ER signaling contributes to BC tumorigenesis and progression; ER transactivates hundreds of genes promoting BC growth [8], including Myc, cyclin D1, BCL-2, and vascular endothelial growth factor (VEGF) [9,10,11,12].

The functionality of ER in breast cancer highlights hormone therapy as the major treatment for ER + ve BCs. The therapy started following the introduction of tamoxifen, a selective estrogen receptor modulator (SERM), in the 1970s [13]. Since then, the selective estrogen receptor degrader (SERD) fulvestrant [14,15] and a set of aromatase inhibitors (AIs) have been developed [16]. AIs are the standard of care in adjuvant-based endocrine therapy in postmenopausal patients [17], while tamoxifen is preferred as adjuvant hormone therapy in the premenopausal setting [13]. Fulvestrant can be used either as first-line hormone therapy or following failure of tamoxifen and AI treatments [18,19]. Endocrine therapy to ER + ve BCs is among the most effective targeted cancer therapies. In first-line tamoxifen treatment, 67% of patients show responses [18,20,21]; adjuvant tamoxifen reduces the annual rate of mortality and recurrence by 31% and 50% respectively [22,23]. The hormonal agents of tamoxifen, fulvestrant, and AIs offer palliation benefits in treating ER + ve metastatic BCs and a long-term delay of disease recurrence under adjuvant setting.

However, endocrine therapy resistance (ETR) occurs. The resistance can be either intrinsic (de novo) or acquired; over 30% of ER + ve BCs display intrinsic ETR [24]. Approximately 30% of patients develop resistance to adjuvant tamoxifen [23,25] and endocrine therapy in general [23,26]. These resistant tumors can remain dormant and are trigged for metastasis up to 20 years following diagnosis [27,28]. For advanced BCs, ETR can be rapidly developed; these tumors are defined as relapse arisen within two years of endocrine therapy or one year after completion of adjuvant hormone therapy according to the recent ESO-ESMO (European School of Oncology-European Society for Medical Oncology) guidelines [29,30]. Relapse of ER + ve BCs to hormone therapy is regulated by a complex network, which includes modulations of ER signaling, activation of epidermal growth factor receptor (EGFR) and other growth factor receptors, NOTCH, microenvironmental cues, and others. However, this rich knowledge at the molecular level has yet to translate into the clinic to counter ETR; the resistance remains a major cause of breast cancer fatality.

At the cellular level, relapse on endocrine therapy requires tumor regrowth, a process that shares similarities with the expansion and regeneration of mammary glands in reproductive cycles of females, in which mammary stem cells (MaSCs) are responsible for the regeneration. Likewise, breast cancer stem cells (BCSCs) mediate the recurrence to endocrine therapy, which is in line with the consensus that cancer stem cells (CSCs) are the driving force of cancer evolution and resistance to therapies [31]. In this regard, the aforementioned molecular mechanisms would be converging on regulating BCSCs and thereby lead to ETR. In this review we will briefly introduce BCSC, review evidence supporting BCSC being a cause of ETR, discuss a strategy to reduce ETR by targeting BCSCs, propose a model of BCSC regulation, and address limitations.

The main materials reviewed in this manuscript were selected according to the PRISMA Guidelines [32,33]. We performed a systemic literature search through the PubMed database using the term: “breast cancer AND endocrine therapy resistance AND breast cancer stem cells”. A total of 94 articles were retrieved; among these, non-English articles (*n* = 3) and not-directly relevant publications (*n* = 28) were removed (Figure 1).

## 2. Breast Cancer Stem Cells

CSCs were first demonstrated in acute myeloid leukemia (AML) as the tumor cells bearing the surface profile of CD34^+^CD38^−^ in 1994 [34]. In solid tumors, BCSCs with the profile of CD44^+^CD24^−/low^ were initially isolated from patients by Al-Hajj et al. in 2003 [35], which ignited enthusiasm in CSC research in solid tumors due to immediate recognition of the central potential of CSCs in cancer initiation, progression, and resistance to therapies. This intensive research has led to the identification of CSCs in almost all types of solid tumors [36,37], including bladder cancer [38], brain tumors [39], colon cancer [40,41], head and neck squamous cell carcinoma [42], liver cancer, lung cancer [43], melanoma [44], pancreatic cancer [45,46], prostate cancer [47] and sarcoma [48]. The initial identification was largely attributed to the assumption of sharing surface biomarkers between CSCs and their counterparts of tissue stem cells (SCs) [49].

The dramatic ability of undergoing cycles of growth and involution throughout the female reproductive life clearly demonstrates the existence of MaSC in the mammary gland epithelial compartment [50]. MaSCs have been demonstrated by the transplantation (mammary gland reconstitution) approach [51,52] and lineage-tracking experiments [53,54,55]. Nonetheless, the identity of MaSCs remains unclear; a variety of marker profiles have been reported in these cells, including Lin^−^CD29^hi^CD24^+^ (Lin^−^: lineage negative) [56], Lgr5^+^Tspan8^hi^ [57], Bcl11b^+^ [58], CD49f^hi^EpCAM^−/low^ [59], and ALDH1^+^ [60]. It is an interesting possibility that these heterogeneous cells, with the ability to reconstitute mammary glands, may represent the breast epithelial hierarchy. For example, quiescent MaSCs replenish proliferative MaSCs, which then generate mammary glands [61]. This similar strategy has been well documented in intestinal epithelial stem cell populations, the quiescent BMI1^+^ stem cells and proliferative Lgr5^+^ stem cells [62]. Intriguingly, BCSCs are also heterogeneous with respect to marker expression. Following the initial identification of BCSCs as CD44^+^CD24^−/low^ESA^+^Lin^−^ (ESA: epithelial specific antigen) [35], ALDH1^+^ [60], CD49f^+^ [63], and CD133^+^ [64] have also been reported to mark BCSCs. The CD133^+^ BCSCs are enriched in TN breast cancers [65], while the CD49f^+^ BCSCs were correlated with resistance to chemotherapy in TN breast cancers [63]. Both CD44^+^CD24^−/low^ and ALDH1^+^ BCSC populations were associated with adverse features in luminal (ER+ve) BCs, including Ki67 as well as basal markers: p63, CK5/6, and CK14 (CK: cytokeratin) [66] and with poor prognosis [67]; ALDH1^+^ BCSCs are particularly correlated with early recurrence following endocrine therapy [68]. These observations support a positive contribution of BCSCs in the development of resistance to endocrine therapy. However, there are differences in the association of these two BCSC populations with features favoring endocrine resistance in luminal BC; the CD44^+^CD24^−/low^ but not the ALDH1^+^ BCSC population was correlated with the HER2 status [66]. The heterogeneous status of BCSCs suggests that individual BCSC populations make unique contributions to ETR in a context-dependent manner. This possibility is supported by the reverse association of the ALDH1^+^ instead of CD44^+^CD24^−/low^ BCSC population with a response to neoadjuvant chemotherapy [69]. The potential contributions of CD44^+^CD24^−/low^ and ALDH1^+^ BCSCs in ETR acquisition will be discussed later.

## 3. Association of Endocrine Therapy Resistance with BCSC Enrichment

Consistent with the CSC model for a central role of CSCs in developing resistance to therapies, cumulative evidence reveals an enrichment of BCSCs in ETR. In vitro, the tamoxifen-resistant derivatives of MCF7 cells (MCF7-TamR) were enriched with BCSCs. The rate of CD44^+^CD24^−/low^ cells in MCF7-TamR was increased [70,71]; elevations in ALDH activity, indicative of an increase in ALDH1 [72], and CD133 [73] were reported in MCF7-TamR compared to MCF7 cells. Treatment with tamoxifen for five days increased the mammosphere forming ability of MCF7 and LM05-E cells (estrogen-dependent murine BC cells) [74]. In vivo, tamoxifen treatment elevated the CD29^hi^CD24^low^ BCSC population [74]. In patients treated with the AI letrozole, an enrichment in CD44^+^CD24^−/low^ BCSCs was observed [75]. Both mammosphere and xenograft tumors derived from patients treated with either tamoxifen or fulvestrant showed increases in ALDH1^+^ BCSCs (Table 1) [76].

Collectively, accumulating evidence obtained from in vitro examination, in vivo research, and clinical samples clearly demonstrates elevations of BCSCs in tumors with acquired resistance to endocrine therapy (Table 1). Intriguingly, the enrichment can be achieved at a very early stage of endocrine treatment. For instance, in vitro treatment with tamoxifen or fulvestrant for five to nine days and in vivo treatment with both agents for 14 days significantly increased the BCSC content [74,76] (Table 1), while prolonged treatment, approximately one-year in vitro and four to five weeks in vivo with tamoxifen, is required to reach a stable status of resistance [77]. These observations thus suggest a causal relationship between BCSC enrichment and the acquisition of ETR.

The BCSCs are enriched in different populations (Table 1). An interesting question is whether these individual populations are largely the same or equivalent. Limited evidence suggests that they are related but not identical. In MCF7 cells, the CD44^hi^CD24^−/low^ and ALDH1^+^ BCSCs were 12.87% and 5.54% respectively with an overlap of 0.084% (Figure 2A) [78]; similar observations were also obtained in MCF10A cells with different percentages [78]. While the CD44^hi^CD24^−/low^ BCSCs display an epithelial-mesenchymal transition (EMT) status, the ALDH1^+^ BCSCs possessed a mesenchymal-epithelial transition (MET) state (Figure 2A) [78]. In a tumor mass, the CD44^hi^CD24^−/low^ BCSCs and the ALDH1^+^ BCSCs are respectively quiescent and proliferative (Figure 2A) [78]. In breast, quiescent MaSCs progress to proliferative MaSCs which then produce mammary glands (Figure 2B) [61]. With this knowledge, it can envisage a model in which CD44^hi^CD24^−/low^ BCSCs are converted to ALDH1^+^ BCSCs and the latter BCSCs directly produce relapsed tumors (Figure 2A). This model can be readily tested. In fact, had the studies (Table 1) examined both populations of BCSCs with respect to their time-wise enrichment, their contributions and relationship to ETR development could be evaluated. Nonetheless, the model (Figure 2A) predicts the generation of ALDH1^+^ BCSCs will be a later event following the early production of CD44^hi^CD24^−/low^ BCSCs in the process of endocrine treatment. This scenario is supported by the inhibition of cell proliferation by tamoxifen and fulvestrant in a short term treatment with the concurrent appearance of CD44^hi^CD24^−/low^ BCSCs [76]; while CD44^hi^CD24^−/low^ BCSCs are quiescent, ALDH1^+^ BCSCs are proliferative (Figure 2A) [78].

## 4. Mechanisms Underlying BCSC Enrichment Following the Development of Endocrine Resistance

Endocrine resistance is regulated by complex mechanisms, including ER, growth factor receptors/PI3K-AKT-mTOR, NOTCH, Wnt, Hippo-YAP/TAZ, and microenvironment cues (Figure 3). We will discuss the impact of these mechanisms on BCSC enrichment in ETR.

### 4.1. The Relationship of ER Signaling and BCSCs in Resistance to Endocrine Therapy

Luminal BCs are largely ER + ve tumors; based on the guidelines of the American Society of Clinical Oncology/College of American Pathologists (ASCO/CAP), luminal BCs are tumors with ≥1% of ER + ve or PR + ve cells [79], indicating the importance of ER in predicting the response to hormone therapy. The primary mechanism for intrinsic resistance to endocrine therapy is a lack of ER expression [23]. In the setting of acquired ETR, loss of ER occurred in 17–28% of relapse tumors [80,81,82] and the ER + status is largely maintained in resistant tumors [81,83,84]. In patient-derived xenograft (PDX, ER + ve) tumors, tamoxifen treatment and estrogen removal, which mimics AI treatment, increased ER content [85], which supports contributions of ER signaling to endocrine resistance. The ER in relapse BCs is functional, as a large proportion of these tumors are sensitive to alternative endocrine treatment. For example, nearly 20% of relapse BCs resistant to tamoxifen respond to second-line endocrine treatment with AIs or fulvestrant [86,87]. Recent evidence favors that ER (ERα) signaling is a major mechanism underlying endocrine resistance [88,89]. Sustaining of ER function can be mediated by modulations of either the ER or its transcriptional co-factors.

#### 4.1.1. Alterations of ER in Relapse Tumors Treated with ENDOCRINE Therapy

Amplification of the ERα encoding gene *ESR1* occurs in preclinical settings involving tamoxifen and fulvestrant [90] and in clinical BCs progressed in endocrine therapy with varied frequency [91,92,93,94,95,96]. The amplification was enriched in bone metastases [97] and could be associated with poor prognosis in patients with ER + ve BCs following adjuvant tamoxifen therapy [95]. *ESR1* amplification was observed in relapse tumors progressed on either AI (24%) or tamoxifen (13%) [98]. Additionally, in both discovery and validation cohorts, CYP19A1 (encoding aromatase) amplification was detected in 16% and 32% of AI-treated patients respectively as well as in 3% and 5% of patients treated with tamoxifen [98]. CYP19A1 amplification promotes ER signaling through estrogen autocrine [98]. In support of this autocrine-mediated ER activation, AI treatment enhances cellular cholesterol biosynthesis, thereby upregulating estrogen autocrine [99].

Accumulating evidence as reviewed by Pejerrey et al. [100] clearly underpins a major contribution of recurrent *ESR1* mutations, particularly in the hormone binding domain (HDB), in developing ETR and specifically in metastases that resulted from these resistant tumors. Ligand-independent ER activation occurred through missense mutations in the *ESR1* gene, including Y537S, Y537N, and D598G [101,102,103,104,105]; these mutations arose in relapse tumors following endocrine therapy and their expression conferred resistance to tamoxifen [103,104,105]. They were detected in nearly 20–30% of relapse BCs [103,104,105,106,107], indicating a major role of these mutations in the development of endocrine resistance. This concept is in accordance with the repeated discovery of these mutants and E380Q in numerous studies (see Review by Pejerrey et al. [100] for details); mutations in *ESR1* arise in 20–40% of metastatic BCs treated with endocrine therapy [100]. Functionally, ectopic expression of the Y537S mutant in MCF7 cells enriched CD44^+^CD24^−^ cells, increased cell’s mammosphere formation capacity, and upregulated stemness genes including OCT4, SOX2, SOX9, BMI1 and others along with an enrichment in NOTCH signaling [108]. The enhancement of BCSCs required ER signaling, as evident by an increase in the phosphorylation of ERα at serine 118 (pS118) and inhibition of this event abolishing BCSC enrichment [108]; pS118 is well characterized to increase ER transcription activity and is induced by tamoxifen [109]. ERα-Y537S was also incapable of enhancing BCSCs upon inhibition of NOTCH signaling, demonstrating that the interplay between ER and NOTCH plays a critical role in ERα-Y537S-mediated BCSC enrichment [108]. Overexpression of either the *ESR1-Y537N* or *ESR1-D598G* mutant in MCF7 also induced pS118 along with an upregulation of mammosphere formation [108]. Collectively, evidence supports that *ESR1* mutations in HDB, at least for Y537S, Y537N and D598G contribute to endocrine resistance at least in part via BCSC enrichment.

Besides the above missense mutations, a set of *ESR1* fusion genes were detected in endocrine resistant tumors. The *ESR1* gene consists of 10 exons consisting of (1) two non-coding exons, (2) exons 3–6 encoding an activating motif, DNA binding domain, and a hinge region, and (3) the remaining exons coding for an LBD (ligand-binding domain) (Figure 4). The first two exons of ESR1 were found to fuse with an N-terminal truncated CCDC170 fragment (ΔCCDC170) (Figure 4) [110]; the fusion gene was enriched in endocrine resistant luminal B breast cancers and its expression was driven by the *ESR1* promoter [110]. Ectopic expression of ΔCCDC170 conferred resistance of T47D cells to tamoxifen in vitro and in vivo (xenografts) [110]. Similar fusion with C6orf211 was also identified in tumors resistant to AI (Figure 4) [111]. Additionally, recurrent fusion genes are more commonly involved in the N-terminal *ESR1* at breakpoints between exons 6 and 7 to C-terminal partners (Figure 4. The *EST1* portion in these fusion genes lacks the LBD (Figure 4); the fusion proteins thus bind ER targets independent of the ligand estrogen. The fusion partners include in-frame C-terminal YAP, PCDH11X [112,113], DAB2 (disabled homolog 2), GYG1 (glycogenin 1), SOX9, MTHFD1L, PLEKHG1, TFG, NKAIN2, AKAP12, and CDK13 (Figure 4) [114]. The fusion proteins of ESR1-YAP1, ESR1-PCDH11X, ESR1-DAB2, ESR1-GYG1, and ESR1-SOX9 substantially increase ER transcriptional activity, independent of ligand association, and confer anti-hormone actions [112,113,114]. While the contributions of these ESR1 function genes in BCSC acquisition during ETR remain to be directly demonstrated, ectopic expression of either ESR1-YAP1 or ESR1-PCDH11X in T47D cells conferred resistance to fulvestrant with concurrent enrichment in the ER and EMT processes [113]. EMT plays a fundamental role in CSC [115]. Evidence thus suggests these fusion genes contribute to endocrine resistance in part through BCSC enhancement.

#### 4.1.2. Elevation in ER Signaling via Its Co-Transcriptional Factors in Endocrine Resistance

The association of ER with chromatin requires its pioneer factors, including FOXA1 [8,116,117], PBX1, and others [118,119]. Amplification of the *PBX1* gene in ER + ve tumors was associated with metastasis and poor prognosis. PBX1 facilitates ER signaling [120], was suggested to control 70% of ER response, is required for ER-mediated proliferation of MCF7 cells, and can stratify the metastatic risk of ER + ve BC [121]. Evidence thus supports an important role of PBX1 in developing endocrine resistance.

In a study of 1501 ER + ve BCs including 692 tumors treated with hormone therapy, genomic alterations in ER transcriptional regulators including FOXA1 were enriched in the treated tumors [122]. FOXA1 was reported to program ER binding to genes functioning in ETR development and BC metastasis [123]. Some targets were AGR2 and IL-8; FOXA1 mediated AGR2 expression in tamoxifen-resistant ER + ve BC cells [124] and promoted resistance to tamoxifen in part through IL-8 functions, as knockdown of IL-8 attenuated the resistance [125].

Besides the pioneer factor to recruit ER to chromatin, Mediator Subunit 1 (MED1) brings a mediator complex to the ER via direct association with the ER, thereby playing an important role in ER-derived target gene expression [126,127]. Consistent with HER2 promoting the endocrine resistance of ER + ve BCs, MED1 expression was correlated with HER2 expression and facilitated the communication between ER and HER2, which contributed to HER2-derived promotion of endocrine resistance; downregulation of MED1 reduced the resistance [128,129,130]. Collectively, evidence supports that through mediating ER target gene expression, the pioneer factors and co-regulators of ER play roles in endocrine resistance.

ER transcriptional activity is regulated by its co-activators with AIB1 (the ER modulator Amplified in Breast Cancer-1) being notably involved in endocrine resistance. *AIB1* amplification and its upregulation in expression were detected in 10% and more than 50% of primary BCs respectively [131]. AIB1 upregulation is associated with HER2 expression and resistance to tamoxifen in primary BCs [132]. AIB1 binds to the ER in response to tamoxifen, which enables the ER to transactivate HER2 [132]; AIB1 thus plays a direct role in bridging the communication of ER with the HER2 growth factor receptor; the communication contributes to BCSC enrichment in endocrine resistance (see Section 4.2). Additionally, AIB1 sustains embryonic stem cells through facilitation of OCT4, NANOG, and SOX2 expression [133], and promotes CSC formation [134]. In ER + ve BCs, AIB1 contributes to the formation of ALDH+ mammospheres [135]. Collectively, evidence supports a role of the ER co-activator AIB1/NCOA3/SRC3 (steroid co-activator 3) in ETR via BCSC enrichment.

#### 4.1.3. The Impact of ER Signaling on Endocrine Resistance-Associated Enrichment of BCSCs

Relapse of ER + ve BCs on endocrine therapy results in tumor regrowth and evolution; both processes are attributable to BCSC’s capacities of tumor initiation and plasticity. Likewise, ER signaling sustained under hormone therapy will be expected to enrich BCSCs. There is indirect evidence supporting this concept. MaSCs are ER-ve and ER signaling reduces the pool of MaSCs [61,136]. Whilst the ER status of BCSCs enriched in ER + ve tumor cells resistant to endocrine therapy is not clear, evidence suggests the ER-ve status of both CD44^+^CD24^−/low^ and ALDH1^+^ BCSCs [76,137,138]. This concept is in accordance with the observations that endocrine treatment with either tamoxifen or fulvestrant reduced the proliferation of ER + ve cells with a concurrent increase of BCSCs [76] and that knockdown of FOXA1 attenuated MCF7 cell proliferation without impact on mammosphere formation [139]. Furthermore, Nasr et al. have recently established a BC cell line from a ER+/PR+/HER2- tumor; the cell line consists of 92% ALDH1^+^ cells and 0.97–5.4% of CD44^+^CD24^−/low^ cells, and OCT4, SOX2, and NANOG were overexpressed, suggesting the line being essentially cells with BCSC properties [140]. Intriguingly, treating these cells with either estrogen, progesterone, or their inhibitors for six months did not affect cell proliferation [140]; however, the ER status in the cell line prior to and after treatments has not been documented [140].

Nonetheless, evidence supports the involvement of ERα transcriptional activity in BCSC enrichment during ETR progression. A MED1 mutant deficient in binding ER not only reduced ER target gene expression but also decreased the cell’s ability to form a mammosphere [141]. Tamoxifen-induced pS118 increases ERα transcriptional activity [109] and predicts resistance to tamoxifen [142]; additionally, ERα pS118 communicates with SOX2, a critical factor regulating BCSCs (see Section 5), in BCSC acquisition [143]. Furthermore, the recent demonstration of the alterations in *ESR1* following endocrine treatment (see Section 4.1.1) and the direct contributions of the *ESR1* mutants Y537S, Y537N and D598G to BCSC enrichment clearly support the critical roles of ERα transcriptional activity in BCSC enrichment during the course of ETR.

However, the involvement of ER in BCSCs is likely complex. Although MaSCs do not express ERα, they are ERβ-positive. ERβ was reported to contribute to BCSC enrichment under endocrine therapy [144]. Treatment of MCF7 cells with 17-β-estradiol increased mammosphere formation with respect to the number and size of spheres along with a significant upregulation of ERα36 [145]; the isoform lacks both AF-1 and AF-2 transactivation domains and a part of LBD [146]. ERα36 was reported to enhance BCSCs, confer resistance to tamoxifen, and promote metastasis [147,148]. ERα36 is upregulated in BCSCs, maintains CD44^+^CD24^−/low^ BCSCs likely via activating the AKT pathway, and contributes to resistance to antiestrogen treatment [149,150]. Evidence thus reveals contributions of signaling events leading to AKT activation in ETR-associated BCSC enrichment.

### 4.2. Growth Factor Signaling Stimulating BCSC Enrichment in Developing ETR

In both preclinical models of resistance to tamoxifen and AI (long-term adaptation to estrogen deprivation), activation of the EGFR, IGFR (insulin-like growth factor receptor), PI3K-AKT-MAPK, and mTOR pathways was the consistently observed theme [151,152,153,154]. The contributions of these pathways in endocrine resistance have been well reviewed [3,25,154,155,156,157,158,159,160]. In this section we will discuss their connections to BCSCs in the context of ETR development.

Evidence favors the direct relevance of these pathways in BCSCs. In a profiling effort of 500 CD44^+^CD24^−/low^ BCSCs isolated from primary tumors using next generation sequencing technology, upregulations of the PI3K pathway, EGFR, and other growth factors were demonstrated [161]. In these BCSCs, the Wnt and NOTCH pathways along with stemness genes LIF and THY1 were also enriched [161]. The CSC markers of CD44, CD133, and ALDH1A3 were overexpressed with concurrent downregulation of CD24 as expected [161]. Additionally, mutations leading to activation of the PI3K-AKT-mTOR pathway were detected in BCSCs isolated from patient tumors [162,163]; among 11 BCSC samples, eight (73%) harbored gain-of-function mutations in the PI3K-AKT pathway [162]. Furthermore, activation of the PI3K-AKT-mTOR pathway occurs in BCSCs associated with tamoxifen-resistant tumors; mRNA expression profiling of mammospheres expanded from patients treated with and without tamoxifen demonstrated activation of the mTOR pathway [164]. Activation of the AKT-mTOR pathway can be achieved by overexpression of miR-125b or downregulation of miR-424; these alterations confer resistance to AI along with an increase of BCSCs [165]. Collectively, evidence, although not substantial, supports a direct role of the PI3K-AKT-mTOR pathway in promoting BCSCs in the course of endocrine resistance progression.

The functions of EGFR/HER2 in BCSCs are consistent with its specific involvement in MaSCs. EGFR/HER2 is expressed in MaSCs [166,167]. In a study of 577 BC patients, HER2 expression was positively correlated with ALDH1^+^ BCSCs [60]; enforced HER2 expression in BC cell lines increased ALDH1^+^ BCSCs, which was blocked by trastuzumab (Herceptin, a monoclonal anti-HER2 antibody) [168]. The axon guidance receptor UNC5A was recently shown to repress ER signaling and the CSC population in MCF7 and T47D cells; its knockdown enhanced both events along with increasing EGFR expression and AKT activities [169]. On the other hand, the retinoblastoma-binding protein 2 (RBP2) conferred tamoxifen resistance in part via activation of the IGF1R-HER2-PI3K-AKT pathway, as inhibition of PIK3 reduced the resistance [170]. Collectively, accumulating evidence clearly outlines an important role of the growth factor receptor (EGFR/HER2 and IGFR) in BCSC enrichment in the development of hormone resistance.

### 4.3. NOTCH Pathway Regulating BCSCs in Endocrine Therapy

The NOTCH signaling pathway plays a critical role in endocrine resistance through promoting BCSC enrichment [171]. The pathway is enriched in CD44^+^CD24^−/low^ BCSCs isolated from patients [161]. NOTCH is required to maintain multiple normal stem cells during development [172]. Its signaling plays a role in maintaining the bipotent progenitors of human mammary glands [173] and is required for the commitment of luminal epithelial cell fate of mouse mammary glands [174]. In the setting of tamoxifen-based hormone therapy, NOTCH4 was upregulated, conferred resistance to tamoxifen, and contributed to the stemness of tamoxifen-resistant MCF7 cells [175,176]. NOTCH4 induces resistance to endocrine therapy, either tamoxifen or fulvestrant-based, in part through sustaining BCSCs. The observed BCSC regulatory activity in these settings was mediated through the association of JAG1 (a NOTCH ligand) with NOTCH4 [76]. In the clinic, a signature of NOTCH4 with its downstream targets HES and HEY predicts poor response and poor prognosis in two-independent ER + ve BC cohorts; in vivo, the acquired tamoxifen-resistance of PDX is reversed along with reductions of BCSCS when NOTCH4 function was inhibited [76]. In line with these observations, NOTCH4 expression correlated with Ki67 expression (cell proliferation) in clinical samples and its inhibition using γ-secretase inhibitor sensitized TD47 cell-derived xenografts to tamoxifen [177]. Additionally, FKBPL (FK506-binding protein like) was very recently reported to reduce ETR via inhibiting the resistance-derived BCSCs; this inhibition was mediated by downregulation of NOTCH4 and its ligand DLL4 [178]. NOTCH actions may also be involved in a transition from more luminal A type tumors to luminal B BCs; the latter is associated with intrinsic resistance to endocrine therapy [179]. This provides additional support for an important role of NOTCH in developing resistance to hormone therapy. Collectively, these studies provide an elegant and convincing demonstration for a critical role of the NOTCH4 signaling in promoting BCSCs under endocrine treatment.

### 4.4. The Wnt, Hedgehog, and Hippo-YAP/TAZ Pathways

Wnt signaling plays essential roles in maintaining tissue stem cells [180,181], cancer stem cells [182,183], and therapy-derived BCSCs [184]. Some key components of the Wnt pathway are upregulated in CD44^+^CD24^−/low^ BCSCs isolated from patients [161]. In comparison to MCF7 cells, MCF7-TamR cells display resistance to tamoxifen with increases in Wnt signaling, proliferation activities, and EMT; all these are reversed following the addition of Wnt inhibitor IWP-2 [185]. SOX2 promotes resistance to tamoxifen through stimulating BCSCs, a process that is at least in part mediated by SOX2-induced Wnt signaling [71,186]. Wnt signaling works downstream of SOX9 in facilitating BCSCs, thereby promoting hormone resistance [187]; miR-190 sensitizes the response to anti-estrogen treatment by inhibiting Wnt signaling via downregulation of SOX9 [188].

The Hedgehog pathway is highly conserved, and plays essential roles during development and in the maintenance of tissue stem cells [189]. The pathway contributes to the self-renewal of CSCs [190] and its components are expressed in ER + ve BCs [191]; these components promote endocrine resistance [192]. The impact of the hedgehog pathway on BC has been thoroughly reviewed recently [191]. While these studies support a role of Hedgehog in BCSC-mediated resistance to endocrine therapy, direct evidence and detailed mechanisms underlying the process are lacking.

The tumor suppression functions of the Hippo core kinases are mainly mediated by inhibition of the transcriptional coactivators YAP (Yes-associated protein 1) and TAZ (transcriptional coactivator with PDZ binding motif) [193]. Elevations in YAP/TAZ functions promote CSCs [194]. High levels of YAP/TAZ expression are associated with reductions in metastasis-free survival in patients with breast cancer and adverse features of the disease [195]; TAZ is essential for sustaining the self-renewal of CD44^+^CD24^−/low^ BCSCs [195]. It is intriguing that the fusion gene ESR1-YAP promotes resistance to endocrine therapy (Figure 4) [112]. The YAP/TAZ transcriptional coactivators cross-talk with the Wnt, Hedgehog, and NOTCH signaling pathways [196], which likely contributes to YAP/TAZ activities in maintaining the BCSC self-renewal potential. YAP/TAZ are able to sense stress and extracellular or microenvironmental signals to promote CSC evolution, thereby playing important roles in metastasis and resistance to therapies [193].

### 4.5. Microenvironment Contributions to BCSC Evolvement Following Endocrine Resistance Development

The microenvironment has a major impact on cancer evolution and the development of therapy resistance and it also influences the acquisition of endocrine resistance. In a murine estrogen-sensitive BC cell line (LM05-E), attachment to laminin upregulates the pluripotent genes SOX2, NANOG, and OCT4, increases mammosphere formation, and importantly induces resistance to tamoxifen through α6 integrin [197].

Extracellular vesicles (EVs) derived from cancer-associated fibroblasts (CAFs) induce endocrine resistance through regulating BCSCs. CAF-produced EVs from patients with hormone resistant metastatic BCs deliver mitochondrial DNA to the BCSCs of dormant hormone resistant BCs, which induces oxidative phosphorylation and subsequently “wakes up” the dormancy [198]. EVs derived from CAFs of patients with hormone resistant BCs can deliver miR-221, which enhances the generation of CD133^+^ BCSCs induced by endocrine therapy, thereby promoting ETR [199]. These processes are in part mediated by miR-221-initiated actions of the inflammatory cytokine IL6-STAT3 pathway [199].

Inflammation is a major contributor to hormone resistance [200,201], which is in part through stimulation of BCSCs. In this regard, gene expression profiling of MCF7-derived BCSCs shows an upregulation of inflammatory cytokines (like IL8) [202]. FOXA1 stimulates tamoxifen resistance in MCF7 cells in part through upregulation of IL8, as downregulation of IL8 abolishes the resistance [125]. In line with IL8, also known as CXCR8 (C-C-C motif ligand 8), functioning through CXCR1/2 [203], CXCR1/2 plays a key role in maintaining BCSCs [204,205,206]. The IL6-STAT3 pathway is activated after inactivation of the RB tumor suppressor and contributes to the self-renewal of BCSCs in the development of endocrine resistance [207]. The pathway maintains BCSCs in this setting in part via regulating the mitochondrial superoxide level [207]. IL33 was also reported to enhance BCSCs along with upregulations of SOX2, NANOG, and OCT4, and it promotes endocrine resistance [208].

### 4.6. Other Factors—PAK4 Stimulating BCSCs in Response to Endocrine Treatment

P21 activating kinase 4 (PAK4) belongs to the PAK family consisting of 6 PAK members functioning downstream of small Rho GTPases Rac and Cdc42 [209]. Elevation in PAK4 expression correlates with adverse characteristics of BC, including tumor size, lymph node involvement, and invasion [210,211,212]. High PAK4 expression stratifies the risk of tamoxifen resistance and poor prognosis of EV + ve BCs and is associated with poor outcome in patients treated with tamoxifen [213,214]. PAK4 expression is significantly upregulated in MCF7 cells resistant to either tamoxifen or fulvestrant along with BCSC enrichment; its knockdown reverses the resistance [214]. PAK4 promotes BCSCs under endocrine treatment in part via increasing ER transcriptional activity [213]. Collectively, evidence supports PAK4 promoting endocrine resistance at least in part via facilitating BCSCs.

## 5. The Involvement of Core Stemness Genes in Regulating BCSCs during ETR Development

A key contribution of BCSCs in acquired ETR can be further demonstrated by the contributions of core stemness genes of stem cells and CSCs, BMI1, NANOG, and SOX2. BMI1 is upregulated in BC and associates with BC progression and poor prognosis [215,216]. Its function in maintaining the self-renewal of tissue stem cells [62,217,218,219,220] and BCSCs (CSCs in general) [221,222] has been well established. BMI1 confers resistance to tamoxifen in MCF7 and T47D cells in vitro; its overexpression coverts tamoxifen to an agonist to stimulate xenograft tumor growth in vivo [77].

Both NANOG and SOX2 are commonly upregulated in ER + ve BC cells in response to hormone treatment [140,197,208]. Knockdown of NANOG increased the sensitivity of MCF7-TamR cells to tamoxifen [223]. SOX2, a member of the SOX (SRY-related HMG-box) family [224], is a well-established factor required for the maintenance of embryonic stem cells, tissue stem cells and CSCs [225,226]. It plays a key role in BCSC enrichment in response to endocrine treatment through a pathway consisting of SOX2-SOX9-Wnt (Figure 5) [71,186,187]. Overexpression of SOX9 is sufficient to render resistance to hormone therapy [227]; its upregulation is via an RUNX2-ER complex, i.e., SOX9 is a target of ER in this setting [227]. Besides Wnt signaling, SOX9 induces FXY3D expression; the latter promotes SOX9 nuclear localization, forming a positive feedback loop (Figure 5) [228]. FXY3D is a member of the family of Na,K-ATPase regulators containing a FXYD domain [229]. FXY3D contributes to Src activation via forming a complex with ER and Src, which plays a role in endocrine resistance (Figure 5) [228]. Src tyrosine kinase activity enhances endocrine resistance [230] in part by regulating ER function via phosphorylation of ER [231]. Collectively, evidence reveals a pathway in which the SOX2-SOX9 axis stimulates Wnt or FXY3D-ER-Src actions to regulate BCSCs (Figure 5), thereby promoting endocrine resistance.

## 6. Strategy of Targeting BCSCs to Overcome Endocrine Resistance

Based on current knowledge of the mechanisms regulating BCSCs in acquired resistance to endocrine therapy, a variety of approaches have been tested to target BCSCs in order to reduce ETR. Resistance to tamoxifen increases BCSCs and decreases c-FLIP (TRAIL inhibitor), leading to sensitization of the BCSCs to TRAIL-induced apoptosis; administration of recombinant TRAIL depletes BCSCs of tamoxifen resistant tumors in vitro and reduces the growth of MCF7-TamR cell-derived xenografts and tamoxifen-resistant PDX in vivo (Table 2) [232]. Pyrvinium pamoate was reported to inhibit Wnt signaling, downregulate key stem cell factors (NANOG, SOX2, and OCT4), attenuate the self-renewal of CD44^+^CD24^−/low^ and ALDH+ BCSCs, and reduce MDA-MB-231 cell-derived xenograft growth (Table 2) [233]. A small molecule antagonist of CXCR1 Reparixin [234] is able to deplete ALDH^+^ BCSCs in vitro and reduces xenograft growth and metastasis in vivo (Table 2) [205]. Consistent with a role of Src in promoting BCSCs under endocrine therapy (Figure 5) [228], combination of a Src inhibitor dasatinib and a BCL2 inhibitor venetoclaz induces apoptosis specifically in BCSCs (Table 2) [235]. Complex approaches to eliminate BCSCs have been investigated and were comprehensively reviewed [236,237,238,239].

The major mechanisms contributing to endocrine resistance include persistent ER signaling in the majority of resistant tumors, activation of the PI3K-AKT-mTOR pathway, and NOTCH signaling (Figure 3). These would be the ideal pathways for intervention in order to reduce endocrine resistance. However, we need to be cautious in targeting these pathways; strategies used should consider the negative cross-talks among these pathways (Figure 3). For example, in breast cancer targeting HER2 activates NOTCH1 [240] and inhibition of AKT and PI3K enhances NOTCH4 signaling (Figure 3) [241]. PI3K inhibition upregulates ER transcriptional activity (Figure 3) [242,243]. Similar reciprocal feedback has been reported between androgen receptor signaling and PI3K in prostate cancer, suggesting a common theme between hormone receptor signaling and PI3K in BC and prostate cancer [244].

## 7. A Dynamic Model of BCSC Regulation in the Settings of Hormone Therapy

In accordance with an essential role of CSCs in cancer progression, CSCs are expected to evolve following cancer progression. As a result, CSCs display heterogeneity. This concept is supported by multiple pieces of evidence, including the intratumoral heterogeneity observed in multiple tumor types [245,246], the production of different types of xenograft tumors from a single cell lineage [247], and genome instability associated with CSCs [248]. Although a set of antigens (CD44, CD24, CD133, ALDH1, and others) have been identified in BCSCs (Table 1), it is highly possible that BCSCs can be negative for these markers. The common markers of CSCs are CD34^+^CD38^−^ for AML, and CD133^+^, CD44^+^, and others for solid tumors, as these cells display higher abilities of tumor ignition in nude and NOD/SCID mice [37]. However, with NOD/SCID/IL2Rγ^−/−^ mice that are more receptive to xenograft formation, cancer cells negative for these antigens initiate tumors with comparable efficiencies as those cells positive for the aforementioned markers [43,249,250,251].

The evolution of CSCs and BCSCs following cancer progression can be attributed to their lineage plasticity. For instance, recent developments favor the association of partial or hybrid EMT (the co-existence of both epithelial properties and mesenchymal characteristics) with CSCs [252]. In the squamous cell carcinoma of hair follicle, tumors with partial EMT display increases in plasticity and are more aggressiveness compared to carcinomas with full EMT [252,253,254]. In prostate cancer, tumor cells with hybrid EMT possess CSC properties and contribute to metastasis in vivo [255] and are correlated with metastasis in patients [256]. In a limited number of breast cancer patients examined (*n* = 11), circulating tumor cells (CTCs) positive for both epithelial and mesenchymal markers were observed [257]. In a later study involving 130 patients with metastatic BC including 68.5% of ER + vc BCs, CTCs positive for ALDH1+ and marked with both cytokeratins (CK8, 18, and 19, epithelial markers) and TWIST1 (a mesenchymal marker) were associated with lung metastasis as well as reductions in OS and progression-free survival [258]. In a primary BC cohort (*n* = 176), tumors positive for E-cadherin and vimentin (including 37% of ER + ve BCs) were correlated with decreases in OS and disease free survival [259]. Collectively, evidence supports a dynamic regulation of CSCs during cancer progression, i.e., their appearance may not be limited to cells expressing certain proteins.

Human BCSCs are heterogeneous with cells marked with CD44^+^CD24^−/low^, AHDH1^+^, CD33^+^ (Table 1). The relationship among these individual BCSC populations remains largely unclear. Nonetheless, there is evidence suggesting that they are not identical at least for CD44^+^CD24^−/low^ and ALDH1^+^ BCSCs (Figure 2A) [78]; this suggests evolution of BCSCs following the course of ETR (Figure 2A). With this knowledge, we can propose a model in which endocrine treatment induces BCSCs via dedifferentiation (Figure 6A); this model suggests a dynamic regulation of BCSC’s stemness following ETR development and is different from the classical CSC model emphasizing the existence of BCSCs that drive ETR acquisition (Figure 6B). This model (Figure 6A) may explain the existence of CSCs in cancer cell lines that have been cultured for decades in the presence of 10% serum. This culture condition is unlikely able to sustain preexisting CSCs. At least for DU145 prostate cancer cell-derived spheres, they proliferated significantly slower in the presence of 10% serum than their non-stem counterparts [260]. The dynamic model (Figure 6A) indicates that the CSC potential instead of CSCs is preserved in cancer cell lines. This model is supported by a recent development [261]. In a mouse colorectal cancer model with CSCs marked with diphtheria toxin receptor (DTR) under control of the Lgr5 promoter, addition of diphtheria toxin ceased tumor growth as a result of ablation of Lgr5^+^ CSCs [262,263]; diphtheria toxin removal reproduced Lgr5^+^ CSCs and resulted in tumor regrowth [263]. This research by de Sousa e Melo et al. strongly suggests for the first time that, at least in colorectal cancer, CSCs can be acquired through dedifferentiation from cancer cells. Nonetheless, the dynamic model (Figure 6A) and dedifferentiation-mediated acquisition of BCSCs during the course of endocrine resistance development does not exclude the possibility that the acquisition is from cancer cells with some intrinsic traits favoring BCSC dedifferentiation. Should this concept hold true, identification of these potential intrinsic properties will significantly advance our understanding of endocrine resistance particularly and cancer progression in general.

## 8. Conclusions

The central properties of BCSCs with respect to their capacity of tumor initiation and plasticity clearly place them at the center for developing resistance to hormone therapy. The resistant tumors will progress to metastasis, the leading cause of cancer death [264,265]. BCSCs play a major role in metastasis, in part owing to their plasticity to make transitions between EMT [266,267] and MET (mesenchymal-epithelial transition) [268,269], which are required for metastasis [31]. In this regard, the central role of BCSCs in the acquisition of ETR will set the stage for the subsequent metastasis. The contributions of BCSCs to endocrine resistance have been intensively investigated; currently we have a rich knowledge on the mechanisms regulating BCSCs following hormone treatment, which cover ER, growth factor receptor/PI3K-AKT-mTOR, NOTCH, Wnt, Hippo-YAP/TAZ, and stromal cues. These individual mechanisms have been investigated to control the resistance. However, translation of this knowledge to patients has been challenging.

A major challenge may lie in the heterogeneous nature of BCSCs and their dynamic regulation during the course of ETR progression (Figure 6A). It is an intriguing concept that CSCs are a property of cancer rather than a specific group of preexisting CSCs [263]. This knowledge implies targeting CSCs, BCSCs in the case of endocrine resistance, should consider the mechanisms and factors leading to CSC conversion.

## Figures and Tables

**Figure 1 cancers-11-01028-f001:**
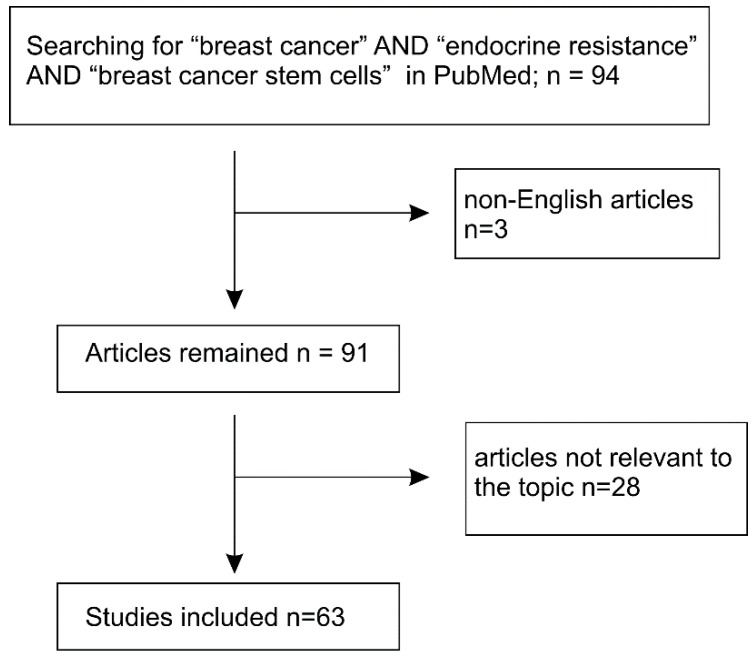
Systemic literature searching conditions and selection of articles for review.

**Figure 2 cancers-11-01028-f002:**
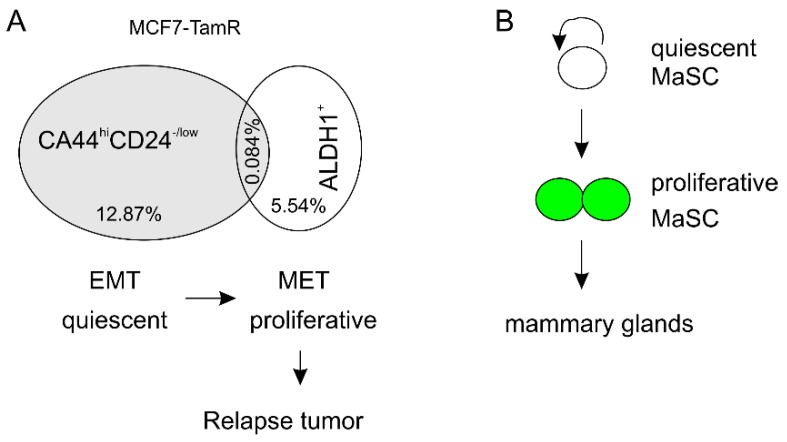
A model shows different contributions of individual BCSC populations in endocrine resistant development. (**A**) The percentage of the indicated BCSC pools in MCF7-TamR cells and their relationship are shown. EMT: epithelial-mesenchymal transition; MET: mesenchymal-epithelial transition. (**B**) Evidence supports that quiescent MaSCs are self-renewal and produce proliferative MaSCs; the latter cells generate mammary glands containing basal and luminal epithelial cells.

**Figure 3 cancers-11-01028-f003:**
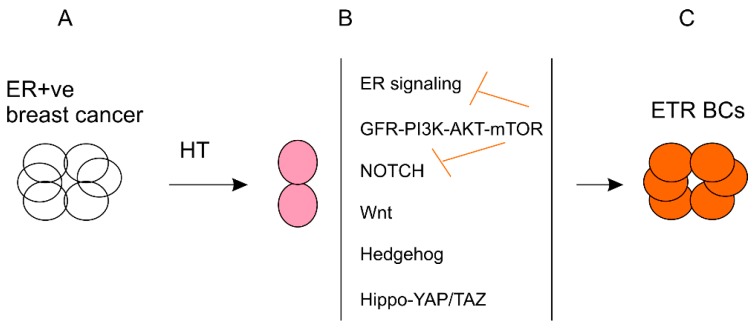
Mechanisms responsible for ETR (endocrine therapy resistance) progression. ER+ve breast cancers are treated with hormone therapy (HT) (**A**). (**B**) The treatment results in surviving cells harboring elevations in ER signaling, growth factor receptor (GFR)-PI3K-AKT-mTOR, NOTCH, or other pathways. The GFR-PI3K-AKT-mTOR pathway inhibits the ER and NOTCH signaling. (**C**) These pathways (**B**) drive the progression of tumors with resistance to endocrine therapy (ETR).

**Figure 4 cancers-11-01028-f004:**
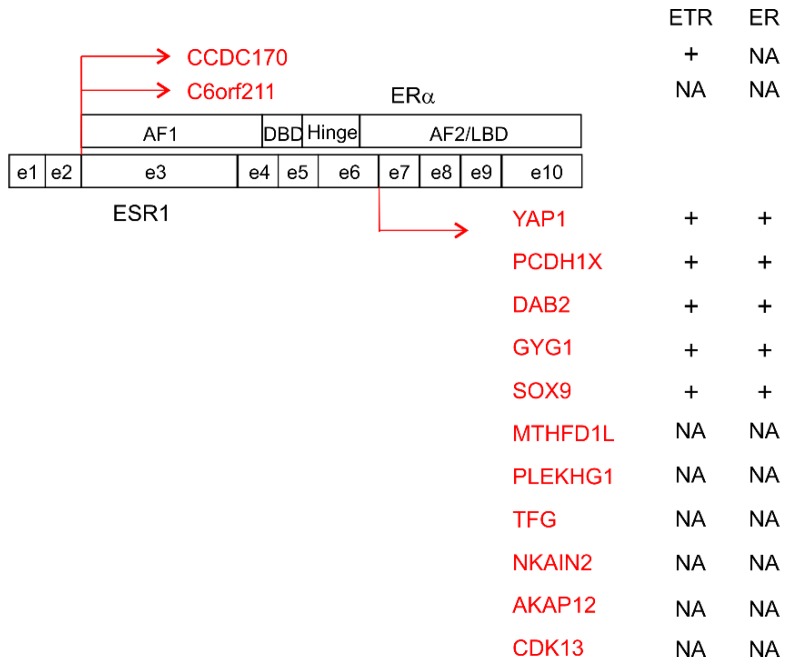
ESR1 fusion genes detected in patients with ETR breast cancers. The exons (e1–e10) of the ESR1 gene and the domain structure of ERα are shown. The fusion of e2 and e6 to the indicated partners and the impact of individual fusion products on ETR and ER transcriptional activity are provided. AF1: activation function 1 domain; DBD: DNA-binding domain; Hinge: hinge region; AF2/LDB: activation function 2/ligand-binding domain; +: enhancement; NA: not available.

**Figure 5 cancers-11-01028-f005:**
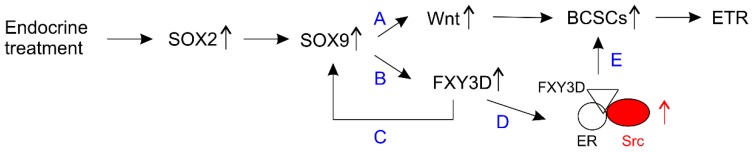
An illustration shows SOX2-initiated events in promoting ETR. Endocrine treatment leads to SOX2 upregulation indicated by the up-pointing arrow; SOX2 subsequently induces SOX9 expression which initiates two processes (**A**,**B**). In (**A**), an increase in Wnt signaling enriches BCSCs which contributes to ETR. In (**B**), upregulation of FXY3D forms a positive feedback loop to enhance SOX9 action (**C**); FXY3D also bridges the formation of the ER-Src complex, resulting in activation of the Src tyrosine kinase (**D**); Src activity facilitates BCSC enrichment (**E**).

**Figure 6 cancers-11-01028-f006:**
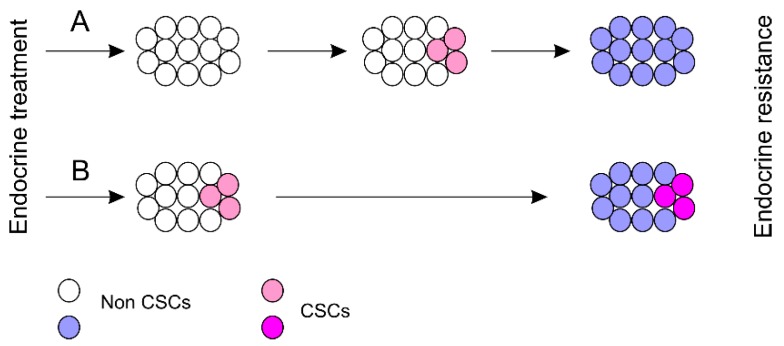
Models of BCSC regulation during ETR acquisition. (**A**) Endocrine treatment results in the acquisition of BCSCs through dedifferentiation (meddle cell population) which then produce tumors resistant to endocrine therapy (right cell population). (**B**) Endocrine therapy induces the pre-existing BCSCs to generate resistant BCs containing a fraction of BCSCs.

**Table 1 cancers-11-01028-t001:** Enrichment of BCSC following endocrine treatment.

Treatment	System ^1^	BCSC ^2^	Ref.
Tamoxifen	MCF7-TamR	CD44^+^CD24^−/low^	[70]
Tamoxifen	MCF7-TamR	CD44^+^CD24^−/low^ Mammosphere	[71]
Tamoxifen	MCF7-TamR	ALDH1^+^	[72]
Tamoxifen	MCF7-TamR	CD133+ Mammosphere	[73]
Tamoxifen	MCF7 and LM05-E, 5 days treatment	Mammosphere	[74]
Tamoxifen	MCF7-TamR	CD44^+^CD24^−/low^	[74]
Tamoxifen	LM05-E xenografts ^3^	CD29^hi^CD24^low^	[74]
Letrozole	Patient BCs	CD44^+^CD24^−/low^	[75]
Tamoxifen ^4^ Fulvestrant ^4^	Patient-derived mammosphere, 7–9 days Patient-derived xenograft, 14 days	ALDH1^+^	[76]

^1^ systems and conditions used to derive endocrine resistance; ^2^ BCSC populations; ^3^ murine tumor; ^4^ both cells isolated from patients samples or patient-derived xenografts were treated individually with tamoxifen and fulvestrant.

**Table 2 cancers-11-01028-t002:** Strategy of targeting BCSCs associated with endocrine therapy resistance.

Method	BCSC	Model	Action	Ref.
TRAIL	Tamoxifen ^1^	Xenograft and PDX	Death receptor	[232]
Pyrvinium pamoate	CD44^+^CD24^−/low^ ALDH^+^	Xenograft	Wnt inhibitor	[233]
Reparixin	ALDH^+^	Xenograft	Agonist of CXCR1/2	[205]
Dasatinib + venetoclaz	BCSCs	In vitro	Src inhibitor BCL2 inhibitor	[235]

^1^ BCSCs were derived from tamoxifen resistant tumors.

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
