# Peer review of "The Central Contributions of Breast Cancer Stem Cells in Developing Resistance to Endocrine Therapy in Estrogen Receptor (ER)-Positive Breast Cancer"

_cancers, 2019, doi:10.3390/cancers11071028_

Reviewer 1 Report

Very interesting topic. The authors made a great effort to summarize some of the complex processes involved in endocrine resistance. The list of references used in the manuscript is comprehensive. However, the notion of cancer stem cells (CSC) is still ambiguous. Previously, CSCs were understood to be ER-, basal-like with self-renewal properties, similarly to mammary stem cells. More recently, the notion of cancer cell plasticity revealed the capacity of maintaining both epithelial phenotype and EMT markers. This is probably the case of BCSCs as discussed in the review article. Some sections need to be improved  

Line 206-207: incorrect statement. Autocrine signaling still remains ligand-dependent, the difference being the local production of estrogen (vs endocrine). This sentence can lead to confusion as ligand independent activation refers to ER all mechanism not involving estrogen ( ei: constitutive activation due to mutations, phosphorylation by RTKs…)

Alterations of ESR1 and their role in endocrine resistance have been discussed in paragraph 4.1.1 (from line 243). However, no clear connection between ESR1 alterations and BCSC has been made. For instance, one would wonder whether missense ESR1 mutant cells (that are endocrine resistant) are predominantly ALDH1+.  Based on you model, ALDH1+ cells may divide into CD44high/CD24low (ER-) and ALDH1+ (ER+). In the context of mutant ESR1, it would be interesting to know whether this can apply. This needs further discussion (include supporting references if available).

The cell line established by Nasr et al (line 274-277) was included to support enrichment of BCSC phenotype from ER+ tumors. Unfortunately, no data was provided to support the expression of ER after the cell line was established. If this is not the case, please provide the correct reference. Nonetheless, possibilities exist that ER expression was lost or significantly decreased during the cell line generation, which may explain the lack of response to ER ligands.  Alternatively, ER can be mutated in the generated cell line. Although ER+ BCSC are not fully dependent ER signaling, estrogen should confer them some growth advantages.

The notion of de-differentiation caused by endocrine therapy is interesting, but direct evidence are still missing (Fig 6). Considering that an acquired resistance usually follows a phase on response, can we assume that the recurrent cells did not have any particular intrinsic trait that allowed them overcome the endocrine therapy? The hypothesis might hold true in a context dependent manner, but it will be too simplistic to generalize it to the entire process of endocrine resistance.

Author Response

We thank reviewer #1 for the insightful comments. Check for spelling errors has been performed; our point-by-point responses are detailed below

“Very interesting topic. The authors made a great effort to summarize some of the complex processes involved in endocrine resistance. The list of references used in the manuscript is comprehensive. However, the notion of cancer stem cells (CSC) is still ambiguous. Previously, CSCs were understood to be ER-, basal-like with self-renewal properties, similarly to mammary stem cells. More recently, the notion of cancer cell plasticity revealed the capacity of maintaining both epithelial phenotype and EMT markers. This is probably the case of BCSCs as discussed in the review article. Some sections need to be improved”

Authors' response – We thank the reviewer for pointing out the recent development on the association of partial EMT with CSCs including BCSCs; this is indeed an important discovery in cancer research, which further outlines the preservation of plasticity in BCSCs. With this consideration, we have added a new paragraph to section 7 “A dynamic model of BCSC regulation in the settings of hormone therapy” (page 13, lines 512-527, marked with red).

“Line 206-207: incorrect statement. Autocrine signaling still remains ligand-dependent, the difference being the local production of estrogen (vs endocrine). This sentence can lead to confusion as ligand independent activation refers to ER all mechanism not involving estrogen ( ei: constitutive activation due to mutations, phosphorylation by RTKs…)”

Authors' response – We agree. The statement has been rephrased (lines 207-208, marked with red), which should have cleared the confusion.

“Alterations of ESR1 and their role in endocrine resistance have been discussed in paragraph 4.1.1 (from line 243). However, no clear connection between ESR1 alterations and BCSC has been made. For instance, one would wonder whether missense ESR1 mutant cells (that are endocrine resistant) are predominantly ALDH1+.  Based on you model, ALDH1+ cells may divide into CD44high/CD24low (ER-) and ALDH1+ (ER+). In the context of mutant ESR1, it would be interesting to know whether this can apply. This needs further discussion (include supporting references if available).”

Authors' response – We agree with these insightful comments. Efforts have been made to discuss the contributions of the ESR1 missense mutations (section 4.1.1, page 6, lines 220-231, marked with red) and ESR1 fusion genes (section 4.1.1, page 7, lines 248-252, marked with red) in BCSC enrichment during the development of endocrine resistance.

“The cell line established by Nasr et al (line 274-277) was included to support enrichment of BCSC phenotype from ER+ tumors. Unfortunately, no data was provided to support the expression of ER after the cell line was established. If this is not the case, please provide the correct reference. Nonetheless, possibilities exist that ER expression was lost or significantly decreased during the cell line generation, which may explain the lack of response to ER ligands.  Alternatively, ER can be mutated in the generated cell line. Although ER+ BCSC are not fully dependent ER signaling, estrogen should confer them some growth advantages.”

Authors' response – As the reviewer pointed out, section 4.1.3 in the last submission was a little too simple. In this revision, we have thoroughly discussed the evidence supporting the complex actions of the ER (transcription, non-genomic actions, and mutations) in endocrine resistance-related BCSC (section 4.1.3, pages 8-9, lines 305-316 and 323-324, marked with red). We trust the additions make the section more comprehensive.

“The notion of de-differentiation caused by endocrine therapy is interesting, but direct evidence are still missing (Fig 6). Considering that an acquired resistance usually follows a phase on response, can we assume that the recurrent cells did not have any particular intrinsic trait that allowed them overcome the endocrine therapy? The hypothesis might hold true in a context dependent manner, but it will be too simplistic to generalize it to the entire process of endocrine resistance.”

Authors' response – We share these thoughtful remarks. These potential intrinsic traits, as the reviewer suggested, have been addressed (page 13, lines 546-551, marked with red).

Reviewer 2 Report

This is a well-written review that was missing in the field. 

Three minor comments:

In section 4.1.2 maybe mention other ER co-factors involved in resistance

In section 4.1.3, ER in BCSC, expression but not ER activity is fully explored in BCSC

Stemness markers in breast cancer were first identified in non ER+ cancers, thus may be biased against those cancers.

Author Response

We appreciate the reviewer’s encouragement. Check for spelling errors has been performed; here are our responses to the 3 minors raised.

In section 4.1.2 maybe mention other ER co-factors involved in resistance

Authors' response – We have added this missing component to section 4.1.2 in this revision (page 8, lines 280-290, marked with red).

In section 4.1.3, ER in BCSC, expression but not ER activity is fully explored in BCSC

Authors' response – We agree. The contributions of ER transcription activity in BCSC enrichment during endocrine resistance development has been added (page 8-9, lines 307-315, marked with red).

Stemness markers in breast cancer were first identified in non ER+ cancers, thus may be biased against those cancers.

Authors' response – We appreciate this insight. The discussion of partial EMT in BCSC regulation in this revision, added in response to the first comment of Reviewer #1, increases the depth of knowledge related with BCSCs in ER+ BCs (page 13, lines 512-527, marked with red).

Reviewer 3 Report

The review article by David Rodriguez entitled “The central contributions of breast cancer stem cells in developing resistance to endocrine therapy in  ER-positive breast cancer” is well written, concise and complete. It is aimed at highlighting the key role played by breast cancer stem cells to support the acquisition of resistance against hormone therapy. The subject of the manuscript is interesting and it is very important because it sheds light on the mechanisms underlying resistance due to breast cancer stem cells.

I have minor comments:

- The legend of table 1 should be better described and detailed and the English should be revised

- The figure 3 should be graphically improved because in the current form it is very confusing and also the legend should be revised

- Similarly, the figure 5 should be graphically improved because in the current form it is very confusing and also the legend should be better detailed

Personally, I think that the manuscript is suitable for publication and that Cancers would be an appropriate place for it to be published.

Author Response

We are grateful to the reviewer’s positive remarks. Spelling and grammar errors have been corrected. Below are our revisions in response to the reviewer’s 3 minors

“The legend of table 1 should be better described and detailed and the English should be revised”

Authors' response – We have revised the Table 1 category and legend (marked with red) and hope this makes the table clearer.

“The figure 3 should be graphically improved because in the current form it is very confusing and also the legend should be revised”

“Similarly, the figure 5 should be graphically improved because in the current form it is very confusing and also the legend should be better detailed”

Authors' response – Both figures have been revised and detailed legends have been included. We trust the revision will make both illustrations easier to follow, to which we thank the reviewer’s comments.